# Cancer Patients’ Perspectives and Requirements of Digital Health Technologies: A Scoping Literature Review

**DOI:** 10.3390/cancers16132293

**Published:** 2024-06-21

**Authors:** Ioulietta Lazarou, Anna-Maria Krooupa, Spiros Nikolopoulos, Lazaros Apostolidis, Nikos Sarris, Symeon Papadopoulos, Ioannis Kompatsiaris

**Affiliations:** Information Technologies Institute (ITI), Centre for Research and Technology Hellas (CERTH), 6th km Charilaou-Thermi Road, P.O. Box 6036, 57001 Thessaloniki, Greece; annamariakrooupa@iti.gr (A.-M.K.); nikolopo@iti.gr (S.N.); laaposto@iti.gr (L.A.); nsarris@iti.gr (N.S.); papadop@iti.gr (S.P.); ikom@iti.gr (I.K.)

**Keywords:** user requirements, cancer patients, scoping review, digital health technologies

## Abstract

**Simple Summary:**

Digital health technologies can help manage the growing cancer burden, but understanding patients’ needs is crucial for these tools to be effective. Our study reviewed existing research on what cancer patients want from digital health technologies. We analysed 128 studies, focusing on web-based platforms, mobile apps, and wearable devices used in cancer care. Patients highlighted the importance of these technologies being easy to use, effective in managing their care, and enhancing communication with healthcare providers. Our findings offer insights for future research to develop digital health tools that meet cancer patients’ preferences, potentially improving their healthcare experience and outcomes.

**Abstract:**

Digital health technologies have the potential to alleviate the increasing cancer burden. Incorporating patients’ perspectives on digital health tools has been identified as a critical determinant for their successful uptake in cancer care. The main objective of this scoping review was to provide an overview of the existing evidence on cancer patients’ perspectives and requirements for patient-facing digital health technologies. Three databases (CINAHL, MEDLINE, Science Direct) were searched and 128 studies were identified as eligible for inclusion. Web-based software/platforms, mobile or smartphone devices/applications, and remote sensing/wearable technologies employed for the delivery of interventions and patient monitoring were the most frequently employed technologies in cancer care. The abilities of digital tools to enable care management, user-friendliness, and facilitate patient–clinician interactions were the technological requirements predominantly considered as important by cancer patients. The findings from this review provide evidence that could inform future research on technology-associated parameters influencing cancer patients’ decisions regarding the uptake and adoption of patient-facing digital health technologies.

## 1. Introduction

Cancer constitutes a global health issue with 19.3 million new cases occurring in 2020, while cancer incidence is projected to significantly increase in the coming years, reaching 28.4 million new cases annually by 2040 [1]. Meanwhile, advances in early detection and cancer treatment have improved survival rates, leading to steadily rising numbers of long-term cancer survivors [2]. The growing population of people diagnosed and living with cancer together with the complex healthcare needs that this population faces place a substantial strain on health services, often resulting in cancer patients and their caregivers not accessing or receiving timely and adequate care and support [3]. Emerging digital health technologies have the potential to contribute towards alleviating the increasing cancer burden exerted on healthcare services by leveraging technology to improve care quality, accessibility, and cost-effectiveness [4].

Digital health technologies are digital products, encompassing both hardware and software solutions and services, for healthcare and related uses intended to benefit people and the wider health and social care system [5,6]. Such technologies may comprise smartphone applications, electronic medical records, wearable monitoring or reporting devices, decision support systems, and online tools for treating or diagnosing conditions, preventing ill health, or for improving system efficiencies [6,7]. Numerous beneficial applications of digital health technologies to healthcare delivery and practice have been reported, including the remote monitoring of patients’ symptoms and conditions; the provision of electronic decision support, resources, and interventions; and the facilitation of distant patient–clinician communication [7,8]. Based on these, the incorporation of digital health technologies into national health systems worldwide has been identified as a key priority [9].

In cancer care, where patients’ symptom burden tends to be high due to disease progression or treatment-associated side-effects, the use of digital health technologies can facilitate the collection of patient-generated data, including patient-reported outcomes. This can help in overcoming the challenges associated with conventional clinician-led symptom monitoring, which can often lead to the under-reporting of patients’ symptoms [10]. Routine symptom assessment enabled by digital health technologies thus has the potential to improve symptom management, resulting, in turn, in improvements in healthcare resource utilisation and patients’ quality of life compared with standard clinical assessment [10]. A further advantage of digital technology applications is that they can increase the equity of cancer care delivery by extending access to care for patients in remote or rural areas or those living with socioeconomic disadvantages, mitigating, therefore, geographical and socioeconomic disparities in cancer outcomes that are associated with limited access to care [11,12]. Moreover, using digital health technologies to support patient education has the potential to improve compliance with care pathways and aid self-management to address the complex healthcare needs of people diagnosed with cancer [7,13]. Despite these advantages, however, the adoption of digital health technologies remains limited both in healthcare overall and in cancer care in particular [10,14].

Incorporating patients’ perspectives into the development of digital health tools has been identified as a critical determinant for the successful uptake of these technologies in healthcare [15]. Understanding end-users’ needs and preferences for technological innovations has been found to increase their acceptance, feasibility, and long-term use in clinical populations [16,17]. To this end, the active engagement of patients and other key stakeholders throughout all stages of technology development via the use of participatory and iterative approaches is recommended and has been shown to be particularly useful for the development of technologies intended to be used by patients with ongoing healthcare and educational needs, such as those experienced by patients diagnosed with cancer [18,19,20]. 

In spite of the reported added value of encompassing end-users’ perspectives into the development of technological innovations, there is limited information available to date regarding cancer patients’ needs and perceptions of digital health technologies, while existing evidence has not yet been systematically reviewed and synthesised. The purpose of the present review, therefore, is to provide an overview of the available evidence regarding cancer patients’ requirements for patient-facing digital health technologies, aiming to enhance our understanding of the needs and preferences of this population for the technologies integrated into their care. This comprehension can contribute towards guiding the development of patient-facing digital technologies to ensure that they meet the expectations of cancer patients and do not place additional burdens on their care, thereby improving their uptake and continued use.

## 2. Materials and Methods

### 2.1. Review Question and Objectives

The main objective of the scoping review was to systematically map and synthesise the existing evidence on cancer patients’ perspectives and requirements for patient-facing digital health technologies, i.e., digital technologies with which patients interact to participate in healthcare or clinical activities [21]. To form the review question, the Population, Intervention, Comparison, and Outcomes (PICO) framework was adopted (Table 1). The review was registered with Review Registry (ID: reviewregistry1816).

### 2.2. Search Strategy

The search strategy comprised two steps. A database, MEDLINE (via PubMed), was initially searched to identify primary research studies reporting on cancer patients’ perceptions of digital health technologies. Relevant text words contained in the title, abstract, and authors’ keywords of identified papers and database index terms were compiled to produce a list of search terms. Three databases were then searched, CINAHL, MEDLINE, and Science Direct, using a combination of subject headings and free-text terms for cancer patients, digital health technologies, and user perceptions (see Figure 1 for the search strategy used for MEDLINE).

### 2.3. Eligibility Criteria

Articles were included if they met all the following a priori specified criteria:Primary research studies.Full-text research articles.English-language publications.Studies conducted with adult (>18) cancer patients.Studies reporting cancer patients’ perspectives on patient-facing digital technologies.

Papers were excluded if they were reporting on the following:

Non-primary studies, including systematic reviews.Opinion articles, editorials, or book chapters.Case-report studies and studies providing no information about sample size.Non-English-language publications.Studies with non-adult (<18) cancer patients.Studies reporting the use of provider-facing digital health technologies.Studies reporting clinicians’/caregivers’ perceptions of the use of patient-facing digital technologies.

### 2.4. Selection Procedure

The study selection process involved three phases. First, all records yielded by the database searches were reviewed for eligibility by title. Then, a selection process based on abstract information was conducted. Finally, a full-text review of studies that met the inclusion criteria according to the initial two phases was performed. Two reviewers (IL, AMK) independently assessed the obtained records for eligibility. Discrepancies at each stage of study selection were resolved through discussion. Finally, citation chaining (forward and backward) was undertaken to ensure that all relevant publications were identified and included, if eligible.

### 2.5. Data Extraction

Two reviewers (IL, AMK) independently extracted relevant data on each included study, the technologies used, and user perceptions using a standardised form developed for the purposes of this review. For each study, the following information was extracted and entered into the form: author(s), date of publication, study design, study population, sample size, technology used, and purpose of technology use. Data on users’ perceptions of technologies were also extracted using the same form and were categorised based on a modified version of the technology evaluation categories of the Human, Organization and Technology-fit (HOT-fit) evaluation framework for Health Information Systems [22]. These categories included perceptions relating to the quality of the technological system, its content/available information, its service quality, and other relevant characteristics and features reported by cancer patients. The results of data extraction were compared and any disagreements between reviewers were resolved by consensus.

## 3. Results

### 3.1. Search Results

Database searches yielded a total of 7836 records. After duplicates were removed, 7793 studies were screened based on their titles, and 709 studies were screened on the basis of their abstracts. Following the screening of titles and abstracts, 391 potentially eligible articles remained, which were examined in full. Of these studies, 107 met the criteria for inclusion. A further 21 eligible studies were identified through forward and backward citation chaining, resulting in 128 included studies (see Figure 2 or the PRISMA flow diagram of the study selection process [23]).

### 3.2. Description of Included Studies

Overall, the included studies were published within the last 20 years (2002–2022), with the overwhelming majority (n = 110, 85.9%) being published from 2013 onwards (Figure 3). Studies were predominantly conducted in the USA (n = 53), Australia (n = 15), the UK (n = 12), Canada (n = 11), and the Netherlands (n = 9). Different research designs were employed by the included studies, with most (n = 55) being interventional, predominantly full-scale or pilot randomised controlled trials (RCTs), or small-scale observational investigations nested within interventional studies (e.g., an evaluation study nested within an RCT). Data collection in all studies was conducted prospectively. Sample size varied greatly between studies, ranging from 3 to 4737 participants, reflecting the diverse objectives and research designs employed by the included studies. The participant population mostly consisted of breast (n = 38), prostate (n = 14), lung (n = 8), colorectal (n = 8), and head and neck (n = 7) cancer patients receiving or awaiting treatment and survivors. The remaining studies (n = 46) included patients with various cancer types or did not provide type-specific information.

### 3.3. Description of Identified Patient-Facing Digital Technologies

Patient-facing digital technologies identified across the included studies can be broadly categorised as follows: Web-based software/platforms (n = 53), mobile/smartphone devices/applications (n = 33), telephone-based services and tools (n = 13), and remote sensing and wearable technologies (n = 4) [24,25,26,27]. Telephone-based services and tools were mostly identified in studies published before 2014, while newer studies (published between 2015 and 2022) predominantly reported the use of Web- and smartphone-based technologies or remote sensing devices. Sixteen studies reported using more than one patient-facing digital technology, mostly including wearable devices combined with additional technologies such as smartphone applications or Web-based platforms (n = 13). In two studies [28,29], an existing social media platform (i.e., Facebook) was used together with other digital health technologies. Identified wearable technologies in all cases comprised activity-tracking devices which were predominantly wrist-worn (n = 15). One study included an upper-arm wearable sensor [24], whist in another study participants were asked to use an accelerometer on their waist [30]. Other technologies, not classified in the abovementioned categories, were reported in nine studies. These included automated chatbots simulating human conversations through text or voice interactions (n = 3) [31,32,33], a touch-screen computer monitor for self-reporting psychosocial information in an ambulatory cancer clinic (n = 1) [34], an electronic health system facilitating patient-reported outcome data collection (n = 1) [35], a virtual reality platform supporting post-treatment rehabilitation (n = 1) [36], a voice-activated, interactive computer model enabling virtual conversations (n = 1) [37], a non-invasive optical technology for the remote screening of severe neutropenia (n = 1) [38], and a telemonitoring system consisting of a small point-of-care haematology analyser linked to a telecommunication hub facilitating the self-testing of patients’ blood counts (n = 1) [39]. Figure 4 presents the number of included studies by technology category.

The majority of the reported technologies (n = 77) were predominantly used for delivering interventions to cancer patients at various stages of disease progression (Figure 5). Of these, 23 provided interventions aiming to assist the self-management of patients’ symptoms and/or overall conditions. In a further 15 studies, interventions were designed to increase patients’ cancer-related knowledge, mostly by providing information and resources on different treatment and care management options. Technologies were particularly used for educating patients on clinical trial participation in one study [40], whilst in another study cancer-related educational materials were developed both for patients and their partners, thus delivering a couple-focused intervention [41]. Sixteen studies described using digital health technology-assisted interventions for promoting patients’ physical activity and health behaviour habits, while a further twelve studies reported delivering therapeutic or educational interventions for the management of psychosocial symptoms. Assisting patient rehabilitation after surgery or chemotherapy was the focus of technological interventions in five of the included studies in this category, whereas an equal number of studies (n = 5) delivered interventions aimed at improving cancer patients’ medication adherence. One study reported the use of speech-generating devices to support telephone or face-to-face communication after total laryngectomy [42]. 

Other intended uses of digital health technologies described in the identified studies included the reporting and/or monitoring of patients’ symptoms and physiological parameters and enabling patient–clinician or peer-to-peer communication. Specifically, the remote reporting and/or monitoring of patients’ conditions, vital signs, and symptoms conducted through self-reported or automatically collected data via wearable technologies was described as the primary intent of digital health technology use in 39 studies. Similarly, the principal purpose of technology employment was the facilitation of remote patient–clinician consultations or peer interactions in nine and three of the included studies, respectively.

### 3.4. Evaluation of Digital Health Technologies

Cancer patients’ perceptions of digital health technologies were almost exclusively investigated by the included studies in the context of evaluations of existing or newly developed patient-facing technologies/technological interventions following their development (n = 115) (Table 2). Only a small number of studies (n = 13) explored patients’ perspectives in the design and/or development phases of technologies. Assessment objectives differed significantly between studies, with 68 papers reporting two or more objectives. Among these, the most frequently evaluated were user satisfaction (n = 43) and the acceptability (n = 41), feasibility (n = 38), and utility/usability (n = 31) of the technologies. Twenty-three studies described exploring cancer patients’ perceptions and/or experiences of using digital health technologies, while the main objective in five studies was the examination of patients’ technological needs and requirements. Despite a plethora of evaluation objectives being described across the studies, there was a significant overlap in the definitions of assessed concepts, with the terms “acceptability”, “satisfaction”, “utility/usability”, and “user perceptions/experiences” being used interchangeably in a number of studies

In terms of the methods employed for the evaluation of identified technologies, 68 studies reported using mixed (i.e., both qualitative and quantitative) or multiple assessment methods, while in 60 studies a single quantitative or qualitative method was utilised. Patient-reported measures, predominantly questionnaires (n = 68), comprised the most frequently described means of technology evaluation, followed by different metrics, such as technology interaction duration and rates, login data, and completion rates, collected either through participants’ self-reports or extracted from technology tracking data (n = 42). Qualitative methods, and particularly individual or group interviews and textual data provided by patients, were utilised in 32 studies. In most cases (n = 62), evaluation methods (questionnaires, scales, interview guides) were developed or adapted/modified from measures and guides used in previous research for the purposes of individual studies without any prior validation (i.e., ad hoc methods). Thirty-two studies described using validated evaluation measures and existing theoretical frameworks for the development of interview guides. Of these, the most commonly employed were the System Usability Scale (n = 10) [43], the Mobile App Rating Scale (MARS; n = 4) [44], and the Technology Acceptance Model (TAM; n = 3) [45].

The included studies overwhelmingly reported positive findings for the patient-facing technology appraisals performed by cancer patients, with 120 studies describing the evaluated technologies being perceived as acceptable, feasible, and/or useful to patients overall. Only eight studies reported mixed findings, concluding with suggested improvements in distinct technological features spanning from content changes to hardware adjustments to increase the usability and acceptance of technologies [25,46,47,48,49,50,51,52]. 

### 3.5. User Perspectives and Requirements

In line with the HOT-fit evaluation framework for Health Information Systems [22], cancer patients’ perceptions of digital health technologies were grouped into four main categories: system, information/content, service, and other requirements. System requirements concern the inherent features of digital health technologies, including the system performance and user interface. Information/content requirements relate to the characteristics of the information delivered to patients through health technologies. Service requirements refer to the extent to which support is provided to patients in the use and handling of technologies. Other requirements cover any additional issues and perspectives raised by cancer patients, not covered by the aforementioned categories. Table 3 and Figure 6 provide an overview of the identified user perceptions and requirements.

In the system requirements category, ease of use (n = 70) and overall system performance (n = 28) were the technological characteristics most frequently assessed as important by cancer patients. Other system features considered by patients included system reliability in terms of the stability of different features (n = 15) and flexibility (n = 15) in being able to return to or skip certain parts or modules and alter their responses. The ease of setting up technologies for the first time and the attractiveness of the overall design and/or user interface were reported as important characteristics by participants in 11 and 15 studies, respectively. In 13 studies, cancer patients highlighted the importance of being able to easily learn how to operate technologies, whilst the system ensuring that provided patient data are secure and protected was reported as a digital health technology requirement in 8 studies. 

In the information/content category, the extent to which the content of the technological intervention was considered useful for the management of patients’ symptoms or overall conditions was the most frequently reported feature among participants of the included studies (n = 86). Using language that is easy to understand and lay terms was raised as an important content feature by participants in 30 studies, while in 26 studies cancer patients noted that alternative means of information presentation, such as videos, pictures, and graphs, are helpful in aiding comprehensibility. The extent to which the presented content is relevant to the patients’ situations and the comprehensiveness of the provided information in terms of covering all important aspects of a given topic were noted by participants in 23 and 21 studies, respectively. In 18 studies, cancer patients reported finding the content of the technological interventions to be tailored to their own needs and situation, which was considered a feature that increased the usability of technologies, whilst in other studies participants commented on the importance of information clarity (n = 12), quantity (n = 10), and reliability (n = 8). Using culturally and gender-sensitive, as well as inclusive, language was noted as an important content characteristic by cancer patients in five studies. 

For the service requirements, cancer patients mostly reported the need to receive appropriate instructions and/or training (n = 20) in the use of technologies, either in the form of a training manual or one-to-one sessions with people who could guide them through the process of setting up and interacting with the technologies. Additional service requirements included technical support being available for the duration of interactions with technologies (n = 9) and receiving feedback on whether given tasks were completed in line with the relevant guidance (n = 4).

Further issues reported by cancer patients which could not be grouped in the aforementioned categories included the technology enabling patient–clinician (n = 50) and peer-to-peer (n = 18) interactions. Specifically, patients highlighted the importance of being able to communicate their symptoms or overall condition with the clinical teams involved in their care through health technologies, either via their data being directly shared with their consulting clinicians or by being able to produce alerts in case of emergencies. Furthermore, in 33 studies participants reported that in the development phase of digital health technologies, consideration should be given to the amount of time that needs to be invested by end-users in order for the appropriate data to be collected, especially in the case of cancer patients where increased symptom burden may not allow for lengthy amounts of time to be devoted to technology interactions. In 18 studies, participants commented that they would appreciate having access to aggregate forms and graphical representations of their own data reported through/gathered from health technologies. Further issues were raised related to the coverage/connectivity of health technologies to internet/telecommunication networks (n = 12), and to the limited battery life of medical devices/wearables (n = 8), deterring patients from using technologies due to additional costs and burden. Lastly, comfort of use was reported as a requirement for technological devices, especially wearables, by patients in nine of the included studies.

**Table 2 cancers-16-02293-t002:** Study characteristics.

Author(s), Year	Country	Study Design	Study Population	n	Technology	Purpose of Technology Use	Technology Description
Abernethy et al., 2009 [53]	USA	Prospective observational study	Metastatic breast cancer patients	66	Tablet-based software	Symptom and quality-of-life reporting	Tablets used by patients for symptom and quality-of-life self-reported assessments
Admiraal et al., 2017 [54]	Netherlands	RCT	Breast cancer patients	138	Web-based platform	Delivery of a tailored psychoeducation programme	Features included assessments of patients’ distress and prevailing problems and psychoeducational materials tailored to their reported needs
Aiello et al., 2006 [55]	USA	RCT	Breast cancer patients	160	Tablet-based software	Enabling electronic data collection on breast cancer risk factors	Passive digitizer tablet computer with a plastic pointer to complete electronic questionnaires through Microsoft Access
Albrecht et al., 2011 [56]	Germany	Mixed methods study	Breast cancer patients	9	Video-based decision aid	Providing guidance in patient decision-making and communicating preferences	Adaptation of existing 55 min video on different surgical treatment options (lumpectomy vs. mastectomy) in the German context using synchronised voice-over
Allenby et al., 2002 [34]	Australia	RCT	Patients with various cancer types	451	Platform accessed through touch-screen computer	Self-reporting of psychosocial information	Touch-screen monitor for patient symptom reporting using software developed utilising the Visual Basic Version 6 programming language; data stored in Microsoft Access 97 databases
Allicock et al., 2021 [30]	USA	Pilot RCT	African American breast cancer survivors	22	Smartphone-based application, waist-worn wearable (ActiGraph wGT3X-BT)	Improving survivors’ physical activity and diet behaviours	App accessed through Samsung Galaxy Core Prime devices used to complete daily ecological momentary assessments; accelerometer for physical activity monitoring
Alshoumr et al., 2021 [57]	Saudi Arabia	Qualitative study	Patients with various cancer types	22	Web-based platform and monitor	Providing patient education, enabling communication with clinicians, aiding self-management of care and symptoms	Inpatient portal presented on 55-inch high-resolution monitors placed in front of a patient’s bed in clinical wards
Appleyard et al., 2021 [58]	UK	Mixed methods observational study	Advanced prostate cancer patients	40	Tablet-based platform	Enabling the electronic collection of patient-reported outcomes	Platform facilitating patient completion of health-related quality-of-life questionnaires on a tablet
Badr et al., 2016 [59]	USA	Qualitative study	Oral cancer survivors	13	Web-based platform	Improving survivor self-management and survivor/caregiver quality of life	Education for the management of side-effects, facilitation of connections with peers, and provision of self-monitoring features
Basch et al., 2007 [60]	USA	Prospective observational study	Cancer patients receiving chemotherapy	180	Web-based platform	Enabling the electronic collection of patient-reported outcomes	Platform features include the real-time report generation of patient symptoms and an alert system for clinicians
Basch et al., 2017 [61]	USA	Feasibility study nested within an RCT	Cancer patients receiving radiation treatment	152	Web-based platform	Enabling patient reporting of symptomatic adverse events	Online patient-reported outcome items for the systematic identification of adverse events in clinical trials
Basch et al., 2020 [62]	USA	Multicentre RCT	Patients with various cancer types	496	Web-based platform	Enabling remote patient symptom monitoring	Online patient-reported outcome items for the systematic identification of adverse events in clinical trials
Beaver et al., 2012 [63]	UK	Exploratory RCT	Colorectal cancer patients	65	Telephone service	Delivery of follow-up consultations	Audio-recorded telephone sessions pre-registered on computerised hospital information systems together with patients’ medical records
Beaver et al., 2009 [64]	UK	RCT	Breast cancer patients	374	Telephone service	Delivery of follow-up consultations	Nurse-led telephone follow-up intervention using a standardised protocol to address various patient concerns, including symptoms, information needs, and psychosocial support, ensuring comprehensive and consistent patient care remotely
Bender et al., 2022 [65]	Canada	Pilot feasibility study	Prostate cancer patients	29	Web-based application	Delivery of peer support for care management	True North PN: Web-based peer navigation programme for prostate cancer patients using a matching algorithm, private messaging, health resources, and case management
Bender et al., 2013 [66]	Canada	Multi-method study	Breast cancer patients	100	Web-based synchronous text message communication platform	Delivery of online support groups	Online communities–websites that offer discussion forums or chat rooms to support breast cancer survivors by providing information, symptom management, and emotional support
Bennett et al., 2016 [67]	USA	Prospective observational study	Patients with various cancer types	112	Web-enabled tablet computer, interactive voice response system (IVRS)	Collection of patient-reported outcomes	IVRS accessed through mobile phones or landlines. Touch-screen tablets provided to users at clinic visits
Benze et al., 2019 [68]	Germany	Prospective feasibility study	Advanced cancer patients	40	Smartphone-based application	Enabling the electronic collection of patient-reported outcomes	MeQoL app for ePROs in advanced cancer patients (symptoms, pain intensity, quality-of-life data)
Bol et al., 2013 [69]	Netherlands	Prospective randomised trial	Older lung cancer patients	357	Website	Delivery of cancer-related information	Personalised audio–visual information in addition to text on website satisfaction and recall of cancer-related online information
Bolle et al., 2016 [70]	Netherlands	Think-aloud study	Older cancer patients and survivors	15	Existing cancer information tools	Providing cancer-related information	3 cancer information websites, 3 Web-based question prompt lists, 1 decision aid based on the values clarification method
Brennan et al., 2022 [71]	Ireland	Prospective feasibility study	Upper gastrointestinal cancer survivors	12	Web-based platform	Delivery of a multidisciplinary rehabilitation programme	Clinically tested digital therapy platform for hosting video calls (one-to-one and group), providing exercise prescription, and appointment scheduling
Cadmus-Bertram et al., 2019 [72]	USA	Pilot RCT	Breast and colorectal cancer survivors	50	Wrist-worn wearable (Fitbit) linked to electronic health record	Monitoring of and education on physical activity	Fitbit activity trackers linked to electronic health records (EHRs), along with email-based coaching and an educational handbook
Chaix et al., 2019 [31]	France	Prospective observational study	Breast cancer patients	4737	Chatbot	Improving medication adherence	The chatbot provides personalised text responses to user questions using machine learning methods
Chee et al., 2017 [73]	USA	Prospective observational study	Asian American breast cancer survivors	5	Web-based platform	Delivery of culturally appropriate patient education	Culturally tailored internet cancer support group for Asian American breast cancer survivors. Pilot test for enhancing women’s breast cancer survivorship experience
Cheville et al., 2018 [74]	USA	RCT	Late-stage cancer patients	516	Web-based platform, pedometer, telephone service	Remote monitoring and management of symptoms and physical activity	Web-based interface and telephone-based IVR used for collection of pain and function patient-reported outcomes; pedometers used for step count
Childes et al., 2017 [42]	USA	Online questionnaire survey	Head and neck cancer patients after total laryngectomy	265	Text-to-speech-based applications/software programmes accessed using various devices	Providing verbal communication support	Speech-generating devices (SGDs) to support telephone or face-to-face communication post-laryngectomy along with text-to-speech for both face-to-face and phone communication and teletypewriter devices as an alternative communication method for individuals post-laryngectomy
Chow et al., 2021 [28]	USA	Pilot RCT	Haematologic malignancy survivors	41	Smartphone-based application, wrist-worn wearable (Fitbit), social media platform (Facebook)	Delivery of an intervention to improve diet and physical activity	mHealth-supported intervention utilising a Fitbit wearable wristband for tracking daily steps, the Healthwatch360 app for monitoring dietary intake (sodium, saturated fats, and added sugars), and a private Facebook peer support group for social interaction and educational support among haematologic malignancy survivors
Chow et al., 2019 [75]	USA	Prospective observational study	Patients receiving active cancer treatment	52	Short Message Service (SMS) over networks of mobile operators, Web-based platform	Enabling distress monitoring	Weekly distress screeners via text message on their personal smartphones, utilising secure links delivered through a Qualtrics SMS Survey tool. The PHQ-4, a validated measure of distress, was completed online, with geolocation data automatically logged upon completion to track screening locations. Automated alerts were triggered for high distress scores
Cleeland et al., 2011 [76]	USA	RCT	Patients following cancer-related thoracotomy	100	Automated telephone service	Postoperative symptom monitoring	Automated telephone calls monitored postoperative symptoms for thoracotomy patients via interactive voice response system. Symptom alerts were sent to clinicians for severe symptoms via email
Collins et al., 2017 [77]	Australia	Pilot RCT	Head and neck cancer patients	30	Web-based platform	Delivering acute symptom monitoring, nutritional management, and swallowing and communication rehabilitation	Participants used PCs, smartphones, or tablets with cameras and microphones to connect with RBWH clinicians via a secure telehealth portal, employing videoconferencing units and webRTC technology. The telehealth system included training sessions for both patients and clinicians, ensuring effective use of the technology for remote consultations
Crafoord et al., 2020 [78]	Sweden	Mixed methods study	Breast and prostate cancer patients	149	Smartphone/tablet-based application	Reducing symptom burden during cancer treatment	App components: symptom self-assessment, evidence-based self-care advice, summaries/graphs of reported symptoms, urgent/persistent system notifications to health professionals
Crawford et al., 2019 [79]	USA	Development and feasibility testing study	Patients with various cancer types	30	Smartphone-based application	Delivering patient education on oral anticancer medications	The app interfaces with patients’ electronic medical records and is designed based on learning style and adherence theories, providing information through text, pictures, animations, and audio voiceovers (adherence barriers and features for dose scheduling, refill reminders, and feedback collection through reflective questions)
Denis et al., 2014 [80]	France	Prospective pilot study	Lung cancer patients	42	Web-based platform	Remote symptom monitoring	Patients self-report symptoms weekly via the internet for lung cancer follow-up—Software tracks weight and 10 symptoms, enabling early relapse detection
Duffecy et al., 2013 [81]	USA	Prospective pilot study	Post-treatment cancer survivors	31	Website	Skill management training for distress education	Components included lessons on basic cognitive behavioural concepts, interactive tools, self-monitoring features, and a discussion board
Eakin et al., 2012 [82]	Australia	RCT	Breast cancer patients	143	Telephone service	Delivery of an exercise intervention	Telephone-delivered intervention with mixed aerobic and resistance exercise guidance to deliver personalised exercise programs and offer remote support, ensuring accessibility for women living in rural areas
Ferguson et al., 2016 [83]	USA	RCT	Breast cancer survivors	47	Videoconferencing equipment and platform	Delivery of Cognitive Behavioural Therapy-based training	Videoconference technology (Tandberg centric 1700 MXP units) to deliver Memory and Attention Adaptation Training (MAAT) and supportive therapy (ST) remotely to breast cancer survivors, improving access to cognitive rehabilitation for participants in various locations
Finlay et al., 2020 [46]	Australia	RCT	Prostate cancer survivors	71	Web-based platform	Delivery of a physical activity intervention	Web-based computer-tailored interventions with different website architectures (free choice vs. tunnelled) to promote physical activity among prostate cancer survivors, with engagement differing based on the navigational structure of the site
Foley et al., 2016 [84]	Ireland	RCT	Breast cancer patients	39	Tablet-based application	Delivery of information on basic breast cancer biology, different treatments, and surgical methods	App for iPad to assess its impact on anxiety levels in patients undergoing surgery for breast cancer
Fu et al., 2016 [85]	USA	Prospective useability study	Breast cancer survivors	30	Web- and smartphone-based platform	Delivery of education on self-care strategies for lymphedema symptom management	TOLF: Web and mobile-based health IT system for lymphedema management—avatar technology for self-care strategies and symptom evaluation
Galiano-Castillo et al., 2016 [86]	Spain	RCT	Breast cancer survivors	81	Web-based platform	Delivery of a tailored exercise programme	Web-based interventions used for cancer survivorship support. Telehealth system with internet-based tailored exercise programme for cancer rehabilitation
Galsky et al., 2017 [87]	USA	Single-arm clinical trial	Prostate cancer patients	15	Videoconferencing platform	Enabling remote patient research-related visits	Platform used on PI’s desktop and patients’ smartphones
Gell et al., 2017 [88]	USA	Single-arm clinical trial	Cancer survivors	24	Wrist-worn wearable (Fitbit), SMS over networks of mobile operators, accelerometer (Actigraph GT3X+)	Supporting physical activity	Intervention includes text messages, Fitbit self-monitoring, and health coaching. Measures physical activity with waist-worn accelerometer and portable GPS
Gilbertson-White et al., 2019 [89]	USA	Mixed methods study	Patients with various cancer types	56	Web-based platform	Enabling symptom self-management	eHealth symptom self-management intervention, OASIS. It features multimedia graphics and uses HTML5 for animations to enhance accessibility for users with limited literacy
Girgis et al., 2017 [35]	Australia	Mixed methods study	Patients with various cancer types	42	Electronic health system	Collecting and utilising patient-reported outcome measures for personalised care	System components included self-report assessments, links to online self-care resources, health professional review, and access to patient reports
Graetz et al., 2018 [90]	USA	Randomised controlled feasibility trial	Ovarian cancer patients	26	Web-based platform	Enabling real-time symptom monitoring	Web-based app for postoperative care in gynaecological oncology patients. Reminders for discharge instructions and symptom monitoring through smartphones, iPads, or Web-enabled devices
Greenway et al., 2022 [36]	UK	Qualitative study	Head and neck cancer patients	7	Virtual reality platform	Enabling access to cancer-related information and resources	WebXR platform ‘recovery’ to simulate a virtual reality experience within a virtual room. This platform allows patients to navigate and interact with targeted resources and specific learning materials related to their cancer journey, featuring natural landscapes and architectural principles to enhance user experience and facilitate self-management of post-treatment recovery needs
Greer et al., 2019 [32]	USA	Pilot RCT	Young adults after cancer treatment	45	Online chatbot	Delivery of a cognitive and behavioural intervention to increase positive emotions	Vivibot delivers positive psychology skills via prewritten material online. Users interact with an automated system through a decision tree structure
Groarke et al., 2021 [91]	Ireland	Nested mixed methods study within an RCT	Cancer survivors with obesity or overweight	36	Wrist-worn wearable (Fitbit), SMS over networks of mobile operators	Delivery of a behaviour-change intervention	mHealth intervention using a Fitbit activity monitor and SMS contact for an 8-week physical activity goal-setting programme. The Fitbit activity monitor provided real-time physical activity feedback, while SMS contact delivered personalised goal setting and behavioural prompts, integrating behaviour-change techniques such as self-monitoring, feedback on behaviour, and goal setting
Gustavell et al., 2020 [92]	Sweden	Prospective observational study	Patients following pancreatic cancer surgery	26	Tablet- and smartphone-based application	Facilitating person-centred care	App features: assessment of self-reported symptoms, risk assessment models for alerts, access to evidence-based self-care, summaries of symptom history
Harless et al., 2009 [37]	USA	Prospective observational study	Breast cancer patients	39	Voice-activated, interactive computer model	Delivery of educational dialogues	Telehealth intervention using Health Buddy System for head and neck cancer (daily education, guidance, and encouragement)
Head et al., 2011 [93]	USA	Mixed methods study	Head and neck cancer patients	44	Telemessaging device (Health Buddy^®^ System)	Providing an interface for patient–healthcare provider communication	Device that attached to land phone line; questions displayed on device screen appliance; responses given by pressing buttons below the screen
Head et al., 2009 [94]	USA	RCT	Head and neck cancer patients	75	Telemessaging device (Health Buddy^®^ System)	Providing an interface for patient–healthcare provider communication	Health Buddy device communicates intervention algorithms for symptom management. Device plugs into telephone line and electrical outlet for operation
Heiney et al., 2012 [95]	USA	Evaluation study nested within an RCT	African American breast cancer patients	39	Teleconferencing platform	Delivery of therapeutic group sessions	Teleconference support group intervention for African American women with breast cancer utilised structured sessions delivered over 8 weeks, with additional boosters. The intervention featured story sharing and coping strategies, facilitated by two experienced African American group therapists using process-focused leadership techniques
Hochstenbach et al., 2016 [96]	Netherlands	Mixed methods study	Cancer outpatients	11	Web- and tablet-based application	Enabling self-management of pain	Mobile app for patients with daily monitoring and graphical feedback and nurses with patient data analysis and decision support
Jacobsen et al., 2022 [24]	Germany	Prospective observational study	Patients with aggressive haematologic malignancies	67	Upper-arm wearable	Vital sign and physical activity monitoring	Wearable including photoplethysmography, temperature probe, and accelerometery sensors; parameters (heart rate, temperature, respiratory rate, physical activity) calculated using proprietary firmware algorithms
Kanera et al., 2016 [97]	Netherlands	PCT process evaluation	Early cancer survivors with various cancer types	23	Web-based portal	Coping with psychosocial issues and promoting a healthy lifestyle	Portal providing self-management training modules covering return to work, fatigue, anxiety and depression, social relationship and intimacy issues, physical activity, diet, and smoking cessation, and a general information module on residual symptoms
Katz et al., 2016 [98]	USA	Prospective pilot study	Patients after pancreatic surgery for any neoplasm	15	Videoconferencing via tablet	Clinical follow-up following surgical procedures	Existing videoconferencing software (Vidyo, Hackensack, NJ, USA) used on Apple iPad tablets provided to patients
Kearney et al., 2006 [99]	UK	Prospective feasibility study	Cancer patients receiving chemotherapy	18	Handheld-computer software	Facilitating patient monitoring and support	Features provided: assessment of self-reported symptoms, pre-defined symptom-scoring algorithm alerting health professionals to the presence of persistent/urgent symptoms, tailored self-care advice, summaries of symptom history
Kelleher et al., 2019 [100]	USA	RCT	Patients with breast, lung, prostate, or colorectal cancer	89	Videoconferencing platform, website	Delivery of behaviour training in pain coping skills	Four 45 min videoconferencing sessions with a therapist, a website that provides patients with training materials and information, social networking, and daily assessments used to personalise videoconferencing sessions
Kenfield et al., 2019 [101]	USA	RCT	Prostate cancer patients	76	Website, wrist-worn wearable (Fitbit), text messages	Facilitate adoption of lifestyle changes	Website included information and recommendations on four topic areas (get active, eat well, stop smoking, and find support); wearable used to track activity levels, text messages used to reinforce adoption, and continued repetition of the recommendations
Kim et al., 2018 [102]	South Korea	RCT	Breast cancer patients	72	Smartphone-based game	Facilitate patient education	3-week programme using typical multiplayer, social network, and platform-based features
Kim et al., 2016 [103]	USA	Qualitative study nested within an RCT	Cancer patients receiving chemotherapy	12	Tablet-based application	Promote active patient engagement and improve care coordination	Personalised social network built around a patient for collaboration with others involved in their care to enable patient-centred health
Kokts-Porietis et al., 2019 [25]	Canada	Qualitative study	Breast cancer survivors	28	Wrist-worm wearable (Polar A360^®^)	Physical activity monitoring	Activity tracker with a built-in heart rate sensor
Kondylakis et al., 2020 [104]	Italy	Prospective pilot study	Breast and prostate cancer patients	135	ICT platform	Aid cancer care and symptom management	Integrated platform containing serious games, psychoemotional monitoring, a personal health system, and a decision support tool
Kubo et al., 2019 [105]	USA	RCT	Cancer patients receiving chemotherapy	82	Website and smartphone-based application	Mindfulness training programme	Commercially available mindfulness programme (Headspace^TM^)
Lamaj et al., 2022 [38]	USA, Spain	Mixed methods observational study	Cancer patients receiving chemotherapy	154	Medical device based on optical imaging (PointCheck)	Neutropenia monitoring	Non-invasive device operating through imaging the blood flowing through the capillaries in the finger
Lambert et al., 2022 [106]	Canada	RCT	Prostate cancer patients	49	Web-based platform	Delivery of a psychosocial and physical activity self-management programme	Platform including five modules: needs assessment, goal setting and action planning, coping planning, sources of support and motivational tools, celebrating successes achieved, and an information library
Lee et al., 2018 [107]	South Korea	Retrospective review of prospectively collected data	Breast cancer survivors	88	Smartphone-based application, wearable	Self-exercise programme provision	Application providing individual aerobic and resistance training programmes provided by physiatrists; pedometer (InBodyBand) for step count
Livingston et al., 2006 [108]	Australia	RCT	Male colorectal and prostate cancer patients	100	Telephone service	Provide support and care information	Phone calls by nurse counsellors to patients on a range of cancer information and management issues
Livingston et al., 2020 [109]	Australia	RCT	Newly diagnosed cancer patients	43	Smartphone/tablet-based application	Provide support and care information	Features include information provision on cancer symptom management and care services, self-report questionnaires, medical appointment diary, and scheduling
Loh et al., 2022 [110]	USA	Prospective pilot study	Older patients with myeloid neoplasms	38	Website, smartphone-based application, wearable device	Home-based individually tailored exercise programme provision and monitoring of physical activity	Web-based clinician dashboard for patient physical activity monitoring, app for self-reporting of physical activity and provision of exercise prescriptions, wearable activity tracker for step count
Lopez et al., 2021 [111]	Canada	Qualitative component of a multi-method study	Cancer survivors	12	Videoconferencing platform	Delivery of cancer rehabilitation programmes	Publicly funded virtual care platform developed by the Ontario Telemedicine Network
Lozano-Lozano et al., 2019 [112]	Spain	Prospective feasibility study	Breast cancer survivors	80	Smartphone-based application	Patient monitoring and provision of feedback on healthy eating and physical activity	Features included self-report questionnaires, notifications on daily energy balance, and recommendations on physical activity and diet
Lucas et al., 2018 [113]	USA	Qualitative component of a mixed methods study	Brain and lung cancer patients	3	Smartphone-based application, wearable device	Remote monitoring of patients’ health status	App for symptom self-reporting; wearable sensor (Mio Alpha Sports Watch) for heart rate and physical activity monitoring
Lyons et al., 2015 [114]	USA	Prospective observational study	Breast cancer survivors	31	Telephone service	Delivery of an intervention to optimise functional recovery	Telephone-delivered sessions on exercise, managing stress, and functioning better at work and home
Ma et al., 2021 [33]	USA	Prospective observational study	Head and neck cancer patients undergoing radiation therapy	84	Automated chatbot	Symptom reporting and self-management education provision	Interactive Web-based communication system
MacDonald et al., 2020 [115]	Canada	Prospective pilot study	Cancer survivors	35	Smartphone-based application, wearable device, telephone service, Web-based platform	Delivery of a rehabilitation and exercise programme	App (Physitrack^®^) providing progressive exercise prescription, wearable (Fitbit) for activity tracking, telephone calls for health coaching, online learning platform comprising self-management e-modules
Mark et al., 2008 [116]	USA	Survey study	Patients with various cancer types	100	Tablet-based platform	Enabling patient symptom screening and reporting	Pen-based e/tablet that operates through a wireless network hosting a platform providing symptom assessment questionnaires
Matthew et al., 2007 [117]	Canada	RCT	Prostate cancer patients	152	Personal digital assistant	Patient health-related quality-of-life monitoring	PDA using stylus for completion of self-reported questionnaires
McCann et al., 2009 [118]	UK	RCT	Breast, lung, and colorectal cancer patients	56	Mobile-based platform (ASyMS©)	Symptom monitoring and management	Platform developed for completion of self-reported symptom questionnaire and input of physiological data, including a risk model for alerting health professionals to the presence of persistent/urgent symptoms
Meropol et al., 2016 [40]	USA	RCT	Patients with various cancer types	623	Web-based application	Education on patient participation in clinical trials	Platform delivering tailored educational content based on assessed patients’ knowledge and attitudinal barriers
Milbury et al., 2022 [119]	USA	Pilot RCT	Female non-small cell lung cancer patients	54	Videoconferencing platform (Zoom)	Mindfulness training or psychoeducation delivery	Platform accessed through patients’ own devices to attend sessions led by specialists certified in mindfulness-based stress reduction
Mirkovic et al., 2014 [47]	Norway	Mixed methods study	Patients with various cancer types	7	Smartphone- tablet-based application	Enabling the management of health-related issues	App developed using οpen-source framework for building cross-platform mobile apps (PhoneGap) containing four features; messaging with health professionals, symptom assessment, symptom management information, forum to connect with peers
Myall et al., 2015 [120]	UK	Qualitative process evaluation of an exploratory RCT	Patients following primary cancer treatment	12	Web-based platform	Self-management of cancer-related fatigue	Platform features included self-reported assessments, educational sessions, and guidance for structured activities
Nguyen et al., 2017 [26]	Australia	Qualitative study	Breast cancer survivors	14	Wearable activity trackers	Increasing physical activity and reducing sedentary behaviour	Six devices used: Fitbit One, Jawbone Up24, Garmin Vivofit2, Garmin Vivosmart, Garmin Vivoactive, and Polar A300; selected for providing a step count function and an associated app
Nguyen et al., 2019 [121]	Netherlands	RCT	Newly diagnosed cancer patients	232	Mode-tailored websites	Patient education as preparation before consultation on diagnosis and treatment	Four different versions of website developed containing the same information, presented in different modalities (via text, images, and/or patient videos)
Nimako et al., 2013 [39]	UK	Prospective pilot study	Oncology patients receiving chemotherapy	10	Telemonitoring system	Blood, body temperature, and symptom monitoring	System consisted of a small point of care haematology analyser, coupled to a telecommunication hub (tele-hub), enabling patients to self-test their own blood count. The tele-hub consists of a touch screen and keypad and acts as the patient interface, communicating results to a server
O’Brien et al., 2020 [122]	Australia	Online survey	Female breast cancer patients	202	Interaction database (IMgateway)	Patient education on complementary and alternative medicine	Database sets out potential interactions between various complementary and alternative medicines and pharmaceutical drugs
Ormel et al., 2018 [123]	Netherlands	Randomised feasibility study	Patients with various cancer types	16	Smartphone-based application	Self-monitoring of physical activity	Existing GPS fitness-tracking app for iOS and Android
Owens et al., 2020 [124]	USA	Qualitative study	African American lung cancer survivors	12	Smartphone-based application	Education on strategies to combat symptoms related to lung cancer	App (Breathe Easier) contains a combination of audio-directed breathing practices, meditations, and yoga exercises demonstrated using instructional text and images of African American and Caucasian adults aged 55 years or older performing various poses
Ownsworth et al., 2022 [125]	Australia	Mixed methods pilot study	Brain tumour patients	8	Videoconferencing platform	Delivery of psychological support	Platform (Metro South telehealth portal) used by health professionals to provide remote consultations to people in their own homes or local health services
Pavic et al., 2020 [126]	Switzerland	Prospective observational study	Palliative cancer patients	30	Smartphone-based application, wearable	Monitoring of patients’ vital signs, physical activity, and symptoms	App consisted of a patient interface providing digital questionnaires to rate subjective pain and distress, a sensor logging module for recording and transmitting signals from smartphone sensors, and an interface to record wearable sensor signals and transmit these to a secured server. Wearable is a sensor-equipped upper-arm bracelet (Biovotion) measuring physiological and activity parameters
Peipert et al., 2021 [127]	USA	RCT	Breast or colorectal cancer patients	65	Interactive platform, touch-screen device	Education provision on cancer care and treatment options	Multimedia touch-screen device for data collection; educational software with the option to select modules of interest
Pope et al., 2019 [29]	USA	Prospective pilot study	Breast cancer survivors	10	Smartphone-based application, social media platform, wearable accelerometer	Monitoring and promoting physical activity and overall health	Commercially available GPS tracking physical activity app (MapMyFitness), Facebook for intervention delivery, Actigraph GT3X+ accelerometer for physical activity assessment
Post et al., 2013 [128]	USA	Pilot RCT	Breast cancer patients undergoing chemotherapy	27	Personal digital assistant	Delivery of educational videos on symptom communication	Low-tech, non-interactive device used for the delivery of race-concordant videos on how to communicate about pain, depression, and/or fatigue
Price and Brunet, 2021 [129]	Canada	Prospective mixed methods study	Young adult cancer survivors	7	Videoconferencing platform	Delivery of a behaviour-change intervention for promoting physical activity and fruit and vegetable consumption	Teleconferencing technology of participants’ choosing (e.g., Skype)
Purdy et al., 2022 [130]	Canada	Prospective feasibility study	Multiple myeloma patients	29	Online application	Delivery of a home exercise programme	Non-commercial app developed by the University of Alberta (HEAL-Me) providing virtually supervised group workouts, independent home workouts, and independent aerobic exercise
Puszkiewicz et al., 2016 [131]	UK	Prospective one-arm, pre–post study	Cancer survivors	11	Smartphone-based application	Promoting physical activity	Commercial app (GAINFitness) for iOS operating platform providing a physical activity programme based on users’ goals, fitness levels, and equipment they have access to
Reilly et al., 2021 [48]	UK	Qualitative component of a pilot RCT	Cutaneous melanoma patients	13	Tablet-based application	Enabling total self-skin examination	App hosted on Android tablets including an individualised digital skin map and the ability to send electronic reports of any skin concerns, including photographs, to a remote Dermatology Nurse Practitioner
Rezaee et al., 2022 [132]	Iran	Prospective observational study	Breast cancer survivors	25	Smartphone-based application	Providing educational content to improve resilience and quality of life	App developed with the component-based programming approach and user interface developed within the Android Studio environment, including features on the calculation of resilience scores, exercise programmes, patient assessments, and experience sharing among peers
Richards et al., 2020 [133]	UK	Mixed methods prospective pilot study	Patients after discharge following cancer-related upper gastrointestinal surgery	40	Web-based platform	Enabling remote symptom monitoring	System components include a patient website, a Web-based symptom-report questionnaire software, and a Web application interface for the secure transfer of data to electronic health records. Algorithms are programmed into self-report scoring system, allowing severity-specific tailored self-management advice to be provided to patients
Rossi et al., 2018 [27]	USA	Prospective observational study	Endometrial cancer survivors	30	Wrist-worn wearable (Fitbit)	Physical activity monitoring	Fitbit Alta™ used for step count
Ruland et al., 2013 [134]	Norway	Questionnaire-based survey nested within an RCT	Breast and prostate cancer patients	103	Web-based application	Illness management support	App developed at the Oslo University Hospital that includes components on symptom assessment, self-management information and activities, information on condition and treatment options, and communication with peers and health professionals
Ruland et al., 2003 [135]	Norway	Pilot RCT	Cancer outpatients	52	Tablet-based application	Provision of patient support on shared decision-making in symptom management	App components include a symptom assessment tool, shared decision-making, and a care-planning component highlighting to clinicians symptoms patients are experiencing
Russell et al., 2019 [136]	Australia	Pilot RCT	Melanoma patients	69	Website	Delivery of mindfulness intervention	Features included short videos and downloadable PDF transcript of the videos, MP3 audio files of guided meditations, general information about meditation
Skrabal Ross et al., 2022 [137]	Australia	Proof-of-concept trial	Cancer patients receiving oral chemotherapy	22	Online SMS gateway, medication adherence monitoring device (MEMS device)	Supporting adherence to oral chemotherapy	SMS consisted of a one-way (no need to reply) message used to provide reminders to take oral chemotherapy and information about the management of side-effects via hyperlinks to portable document format online documents. MEMS consisted of a pill bottle with a cap that contains a microelectronic chip and tracks the date and time the medication bottle cap is opened and, therefore, the assumed dose taken
Smith et al., 2022 [138]	Australia	Qualitative study	Adult cancer patients	13	Various telephone and video telehealth platforms	Delivery of clinical consultations	Telephone and videoconferencing used to facilitate real-time communication between health professionals, patients, and caregivers during the COVID-19 pandemic
Song et al., 2021 [41]	USA	Pilot feasibility study	Prostate cancer patients	62	Web-based programme	Delivery of couple-focused patient education	Programme accessible from patients’ preferred devices (e.g., smartphone, tablet, or computer), including modules about how couples can work effectively as a team, assess and manage prostate cancer treatment-related side-effects and symptoms, and improve healthy behaviours, and a social support feature with post-module assignments, a moderated online forum, meetings with a health educator, and a resource centre
Spoelstra et al., 2016 [139]	USA	RCT	Patients with various cancer types	75	SMS over networks of mobile operators	Promoting adherence to oral anticancer agent medication	Text messages developed according to Social Cognitive Theory using 160 characters or less, delivered through an automated platform storing associated data
Stephen et al., 2014 [140]	Canada	Qualitative study	Cancer patients and survivors	80	Online chat platform	Delivery of professionally led cancer support groups	Synchronous text communication in password-protected chat rooms
Sundberg et al., 2015 [141]	Sweden	Prospective observational study	Prostate cancer patients receiving radiotherapy	9	ICT platform for smartphone use	Aiding the early assessment and management of patient-reported symptoms	Platform components include patient symptom assessments, a risk assessment model based on symptom occurrence and frequency sending alerts to nurses by text message if symptoms are of concern, continuous access to self-care advice related to symptoms and links to relevant websites, and symptom history presented in graphs over time
Suzuki et al., 2016 [49]	Japan	Prospective observational study	Cancer patients receiving radiation therapy	152	Tablet-based software	Delivery of a psychosocial questionnaire	Electronic touch-screen tablet operated via a stylus used for the completion of patient reported outcomes
Valle et al., 2017 [142]	USA	Pilot RCT	African American breast cancer survivors	35	Wireless scale, wearable activity tracker, Web-based platform	Promoting weight gain prevention through self-regulation behaviours	Activity tracker (Withings Pulse) interfaced with a Bluetooth and Wifi-enabled wireless scale (Withings WS-30, Cambridge, MA) and synced data to a single online account accessed through a mobile app or website which contained graphs of weight and physical activity trends
Van Blarigan et al., 2019 [143]	USA	Pilot RCT	Colorectal cancer survivors	42	SMS over networks of mobile operators, wrist-worn wearable (Fitbit)	Promoting increases in physical activity	Daily text messages providing physical activity prompts and recommendations; Fitbit Flex wristband for tracking physical activity, including steps, distance, active minutes, and calories burned
Van der Linden et al., 2021 [144]	Netherlands	RCT	Brain tumour patients	62	Tablet-based application	Delivery of cognitive rehabilitation programme	App containing educational modules on cognitive functions, influences, compensation, attention, planning and control, and memory. In each module, information about cognitive functions is given and compensatory strategies are provided, together with fill-in exercises to practice the strategies
Visser et al., 2018 [50]	Netherlands	RCT	Breast cancer patients	109	Tablet-based applications	Delivery of support group sessions, clinical contact, and illness-specific information	iPad containing existing apps connected through a shared iCloud account for each group (iBooks for educational materials, FaceTime for remote group sessions, contact app including the email addresses of participants, clinical nurse specialist, and the researcher. Calendar app containing dates of the scheduled video sessions)
Vogel et al., 2019 [145]	USA	Pilot RCT	Ovarian cancer patients	104	Smartphone-based application	Provide information on the usefulness of genetic counselling	iOS (Apple) app providing educational materials on genetic counselling and motivational messages, positive feedback, videos, graphics, and triggers to encourage app use
Vogel et al., 2013 [51]	USA	RCT	Ovarian cancer patients	35	Website	Promoting advance care planning	Prototype website developed using Microsoft. NET framework with Ajax to bring together the HTML and CSS at the front end, and Internet Information Services for Microsoft Windows Servers, an SQL database, and SSL encryption at the back end
Walle et al., 2020 [146]	Germany	RCT	Patients with solid tumours undergoing cancer therapy	66	Videoconferencing application	Delivery of follow-up clinical consultations	The Minxli—Arzt via Video Chat application for smartphones was utilised, including features for scheduling encrypted video calls with verified physicians and a chat function with options to upload pictures
Wallwiener et al., 2017 [147]	Germany	Prospective feasibility study	Advanced breast cancer patients	15	Web-based platform	Enabling reporting of patient-reported outcomes	Real-time registry containing molecular data adapted to include patient-reported outcomes
Wan et al., 2021 [52]	Singapore	Development and evaluation study	Patients undergoing elective colorectal cancer surgeries	5	Smartphone-based application	Improving health outcomes for patients and family caregivers	App on BuddyCare platform encompassing the following features: a surgical timeline (29-day perioperative phase where users receive information packages listing important tasks about how to prepare for surgery, postsurgery monitoring, and discharge care), search functionality, introduction to mindfulness-based practices, daily tasks, alerts, reminders, motivational messages
Waterland et al., 2021 [148]	Australia	Impact evaluation study	Patients preparing for major cancer surgery	31	Videoconferencing platform	Delivery of prehabilitation education session	Zoom used for webinar delivery
Weaver et al., 2007 [149]	UK	Prospective observational study	Colon cancer patients	6	Mobile telephone-based software	Symptom monitoring and management	Patients self-reported symptoms using the phone keypad; data were automatically transmitted to a dedicated server. Each patient’s cumulative toxicity chart was displayed both on individualised Web pages (for review by the study nurse) and on the patient’s phone (for information). If incoming readings gave rise to concern, alerts were generated according to criteria stored both on the phone and the server, and appropriate management actions were communicated to patients
Wickline et al., 2022 [150]	USA	Mixed methods study	Advanced ovarian cancer patients	134	Web-based programme	Enabling symptom monitoring and self-management	System where patients can report and track their symptoms, quality-of-life measures, and decision-making preferences during cancer therapy, delivering self-care instructions targeted to reports of moderate–severe symptoms and providing tips on symptom communication with clinicians
Wilkie et al., 2003 [151]	USA	Mixed methods study	Cancer inpatients and outpatients	116	Desktop-based software	Enabling pain assessment	Microsoft^®^ Windows 95/98 personal desktop computer with a touch-screen (Elo™ monitor) was used to complete the electronic version of the McGill Pain Questionnaire
Wu et al., 2021 [152]	UK	Prospective observational study	Patients awaiting cancer treatment	139	Telephone or video calls	Delivery of a prehabilitation education programme	Telephones or videoconferencing platforms used to deliver home-based prehabilitation including personalised training exercises, dietary advice, medical optimisation therapies, and psychological support
Yap et al., 2013 [153]	Singapore	Prospective observational study	Ambulatory cancer patients	60	SMS over networks of mobile operators	Chemotherapy-induced nausea and vomiting management	Series of questions sent via SMS daily for 5 days post-chemotherapy, following a predeveloped clinical algorithm in consultation with clinical pharmacists; each SMS contained several options in which patients were required to respond by selecting the option number that best reflected their symptoms
Zini et al., 2019 [154]	Italy	Prospective pilot study	Head and neck cancer patients undergoing concurrent chemo-radiotherapy	10	Smartphone-based application	Reporting of clinical parameters, quality of life, and symptoms	App running on Android designed to collect patients’ symptoms, clinical parameters, and provide educational materials, daily self-management suggestions, therapy-cost recording, and peer networking and facilitate interactions with clinicians

**Table 3 cancers-16-02293-t003:** Cancer patients’ perceptions of digital health technologies.

	System								Information/Content										Service			Other						
	Ease of Use	Ease of Learning	Ease of Set-Up	System Flexibility	System Reliability	System Security	Performance	Visual Appeal	Usefulness	Relevance	Clarity	Comprehensiveness	Comprehensibility	Reliability	Tailored Information	Information Quantity	Presentation Format	Inclusive Language	Technical Support	Instructions/Training	Feedback/Follow-Up	Comfort of Use	Battery Life/Portability	Coverage/Connectivity	Peer Interaction	Interaction with Clinicians	Time Considerations	Access to Data
Abernethy et al., 2009 [53]	X								X								X					X						
Admiraal et al., 2017 [54]									X										X	X								
Aiello et al., 2006 [55]	X																											X
Albrecht et al., 2011 [56]									X		X	X	X			X	X											
Allenby et al., 2002 [34]	X										X		X							X	X						X	
Allicock et al., 2021 [30]					X		X		X	X						X										X		
Alshoumr et al., 2021 [57]	X				X		X	X	X				X					X		X								
Appleyard et al., 2021 [58]	X												X				X			X								
Badr et al., 2016 [59]	X				X			X	X	X			X	X	X		X					X			X	X		
Basch et al., 2007 [60]	X								X																	X		
Basch et al., 2017 [61]	X								X				X													X		
Basch et al., 2020 [62]	X								X	X			X													X	X	
Beaver et al., 2012 [63]										X		X																
Beaver et al., 2009 [64]									X			X														X		
Bender et al., 2022 [65]	X		X									X							X						X			
Bender et al., 2013 [66]	X		X			X																			X		X	
Bennett et al., 2016 [67]	X						X																X	X				
Benze et al., 2019 [68]	X								X																		X	
Bol et al., 2013 [69]								X					X		X		X											
Bolle et al., 2016 [70]	X	X										X	X		X	X	X			X							X	
Brennan et al., 2022 [71]	X						X		X														X	X	X	X	X	
Cadmus-Bertram et al., 2019 [72]			X						X		X																	
Chaix et al., 2019 [31]	X				X				X		X				X													
Chee et al., 2017 [73]												X					X										X	
Cheville et al., 2018 [74]	X																			X						X		
Childes et al., 2017 [42]					X		X		X														X					
Chow et al., 2021 [28]	X								X			X			X					X	X						X	X
Chow et al., 2019 [75]	X	X				X				X												X				X	X	
Cleeland et al., 2011 [76]	X								X																			
Collins et al., 2017 [77]	X				X				X	X																		
Crafoord et al., 2020 [78]	X	X		X					X	X		X					X									X	X	X
Crawford et al., 2019 [79]							X	X	X		X		X		X		X			X							X	
Denis et al., 2014 [80]	X																									X		
Duffecy et al., 2013 [81]									X																X			
Eakin et al., 2012 [82]									X																	X		
Ferguson et al., 2016 [83]									X																	X		
Finlay et al., 2020 [46]				X					X	X																		
Foley et al., 2016 [84]									X			X																
Fu et al., 2016 [85]	X	X		X					X	X	X	X	X	X			X											
Galiano-Castillo et al., 2016 [86]					X						X		X		X		X											
Galsky et al., 2017 [87]	X																										X	
Gell et al., 2017 [88]	X								X																			X
Gilbertson-White et al., 2019 [89]	X	X					X	X		X	X						X											
Girgis et al., 2017 [35]	X			X					X	X	X	X	X	X	X		X									X	X	
Graetz et al., 2018 [90]									X																	X	X	
Greenway et al., 2022 [36]	X						X	X	X							X	X		X	X								
Greer et al., 2019 [32]									X								X											
Groarke et al., 2021 [91]									X																	X		X
Gustavell et al., 2020 [92]									X					X												X	X	X
Harless et al., 2009 [37]	X		X						X			X																
Head et al., 2011 [93]	X		X						X				X													X		
Head et al., 2009 [94]			X						X				X													X		
Heiney et al., 2012 [95]									X																X			
Hochstenbach et al., 2016 [96]	X	X			X		X	X				X	X								X					X		X
Jacobsen et al., 2022 [24]																						X	X					
Kanera et al., 2016 [97]									X	X			X															
Katz et al., 2016 [98]			X																					X		X		
Kearney et al., 2006 [99]	X								X						X											X		
Kelleher et al., 2019 [100]									X																			
Kenfield et al., 2019 [101]									X			X							X						X	X		
Kim et al., 2018 [102]	X								X				X															
Kim et al., 2016 [103]	X								X															X		X		
Kokts-Porietis et al., 2019 [25]					X				X																			X
Kondylakis et al., 2020 [104]	X						X																					
Kubo et al., 2019 [105]				X					X						X				X						X			
Lamaj et al., 2022 [38]	X	X					X																					
Lambert et al., 2022 [106]	X						X		X	X					X													
Lee et al., 2018 [107]	X				X					X														X				
Livingston et al., 2006 [108]									X					X												X	X	
Livingston et al., 2020 [109]	X	X					X			X																	X	
Loh et al., 2022 [110]				X																					X	X	X	
Lopez et al., 2021 [111]									X			X		X														
Lozano-Lozano et al., 2019 [112]				X			X	X	X	X		X																X
Lucas et al., 2018 [113]	X			X		X	X		X													X	X			X		X
Lyons et al., 2015 [114]									X	X																	X	
Ma et al., 2021 [33]	X						X																			X		
MacDonald et al., 2020 [115]									X			X				X									X	X	X	X
Mark et al., 2008 [116]	X								X																	X		
Matthew et al., 2007 [117]				X			X		X																			
McCann et al., 2009 [118]	X			X					X			X								X						X	X	
Meropol et al., 2016 [40]									X							X	X										X	
Milbury et al., 2022 [119]	X					X			X									X						X	X			
Mirkovic et al., 2014 [47]	X			X			X	X	X								X											
Myall et al., 2015 [120]	X						X		X				X		X			X		X								
Nguyen et al., 2017 [26]	X	X	X		X			X	X								X					X	X		X	X	X	X
Nguyen et al., 2019 [121]								X					X		X													
Nimako et al., 2013 [39]	X									X										X				X		X	X	
O’Brien et al., 2020 [122]	X							X	X											X						X		
Ormel et al., 2018 [123]				X					X																			
Owens et al., 2020 [124]													X					X										X
Ownsworth et al., 2022 [125]			X												X				X					X		X		
Pavic et al., 2020 [126]	X				X	X	X												X			X	X			X		X
Peipert et al., 2021 [127]									X			X																
Pope et al., 2019 [29]							X		X																			X
Post et al., 2013 [128]	X												X													X		
Price and Brunet, 2021 [129]									X								X									X		
Purdy et al., 2022 [130]	X	X							X																	X		
Puszkiewicz et al., 2016 [131]										X				X	X										X			
Reilly et al., 2021 [48]							X		X																	X		
Rezaee et al., 2022 [132]	X	X			X		X	X		X									X									
Richards et al., 2020 [133]					X		X		X	X					X		X			X		X				X		X
Rossi et al., 2018 [27]	X	X	X						X																			
Ruland et al., 2013 [134]									X																X	X		
Ruland et al., 2003 [135]	X	X		X									X							X								
Russell et al., 2019 [136]									X						X		X				X			X				
Skrabal Ross et al., 2022 [137]								X	X															X				
Smith et al., 2022 [138]						X													X					X		X	X	
Song et al., 2021 [41]	X								X																			
Spoelstra et al., 2016 [139]									X																			
Stephen et al., 2014 [140]						X					X			X											X			
Sundberg et al., 2015 [141]	X								X		X		X				X									X	X	X
Suzuki et al., 2016 [49]	X												X							X								
Valle et al., 2017 [142]	X								X																		X	
Van Blarigan et al., 2019 [143]									X																		X	
van der Linden et al., 2021 [144]	X												X			X										X		
Visser et al., 2018 [50]							X																		X	X		
Vogel et al., 2019 [145]	X								X							X	X	X										
Vogel et al., 2013 [51]	X								X	X						X									X			
Walle et al., 2020 [146]	X						X																	X			X	
Wallwiener et al., 2017 [147]	X												X														X	
Wan et al., 2021 [52]							X	X	X							X	X											
Waterland et al., 2021 [148]	X		X																	X								
Weaver et al., 2007 [149]	X								X																	X		
Wickline et al., 2022 [150]	X								X	X			X														X	
Wilkie et al., 2003 [151]	X										X	X	X				X			X						X	X	
Wu et al., 2021 [152]				X					X																X			
Yap et al., 2013 [153]						X			X			X	X													X	X	
Zini et al., 2019 [154]	X			X					X						X					X								

## 4. Discussion

### 4.1. Main Findings

This scoping review aimed to identify and synthesise existing evidence on cancer patients’ perspectives and requirements for patient-facing digital health technologies. We found 128 studies published between 2002 and 2022 reporting on users’ perceptions and evaluations of patient-centred digital technologies intended to be used as part of cancer patients’ care. A significant surge in publications was observed from 2013 onwards (n = 110, 85.9%), suggesting a growing interest and investment in leveraging digital solutions to address cancer care needs. Identified technologies predominantly comprised Web-based software or platforms (n = 53), mobile or smartphone devices and/or applications (n = 33), remote sensing and wearable technologies (n = 17) used either in combination with other technologies (n = 13) or as standalone digital applications (n = 4), and telephone-based services and tools (n = 13). This diverse range of technologies mirrors the broader technological advancements seen in healthcare, where the emphasis on user-centred design and accessibility has propelled the development and deployment of a wide array of digital tools tailored to specific patient populations [155]. The perceived usefulness of technologies for the management of patients’ symptoms and overall conditions (n = 86), their user-friendliness in terms of ease of use (n = 70), their ability to facilitate patient–clinician interactions (n = 50), and the time investment required to operate the technologies (n = 33) were the most frequently reported technological requirements considered by cancer patients.

The delivery of interventions mostly aiming to educate patients on the self-management of their care (n = 77) and the systematic reporting and/or monitoring of patients’ symptoms and physiological parameters (n = 39) constituted the principal purposes of technology employment among the included studies. Similar findings were reported in previously published reviews investigating the applications of digital technologies in patients living with chronic conditions, such as dementia [156] and serious mental illness [157]. This suggests that people with complex and ongoing healthcare needs may have broadly similar requirements and needs for the digital technologies employed to aid their care regardless of their individual diagnoses. 

Limited digital literacy is a common barrier to digital technology adoption among clinical populations [7,15]. Patients need to have the necessary skills to engage with digital tools independently and meaningfully. These skills comprise a patient’s digital literacy [11]. The user-friendliness of technologies (n = 70), receiving appropriate instructions and/or training in setting up and engaging with technologies, and technical support being readily available for the duration of interactions with technologies (n = 9) were identified as key requirements by patients participating in the studies included in this review. These requirements together with end-users’ digital literacy levels ought to be considered by digital technology companies when designing and developing tools for health purposes so that these can accommodate patients with varying digital literacy levels and ensure that all patients can benefit from digital tools [11].

Patients in a large proportion of studies (n = 50) considered the ability of digital technologies to facilitate patient–clinician communication and interaction an important feature of these tools. Previous research has found that the use of digital health interventions can positively influence patient–clinician communication and relationships by enabling patients to feel more comfortable about disclosing information to their clinicians and by reducing the imbalance of power between the patient and clinician [158,159]. For this to be achieved, it is critical that clinicians work collaboratively with patients to establish mutually agreed goals, boundaries, and expectations for technology engagement [160]. Failing to set and adhere to these may lead to the overburdening of clinical staff, patients experiencing disappointment and frustration, and, ultimately, challenged patient–clinician relationships [161,162]. Given than the ethical implications of digital health communication are complex in terms of the boundaries of the patient–clinician relationship and the professional duty of care, there is a need for guidelines on good practice in the use of technology-enabled communication to be established both at the institutional and national levels [158].

In line with previous research, technological features that enable the personalisation of information and care were highly endorsed by cancer patients participating in studies included in the present review [15,163]. In the context of digital technology, personalisation refers to the process of tailoring the user experience of a digital tool to an individual’s specific needs, preferences, behaviours, and characteristics [15]. In this review, the extent to which the content presented through technologies was tailored (n = 18) and relevant (n = 23) to the patients’ individual situations and was perceived as useful for the management of their own care (n = 86) were identified as desirable technological characteristics. The personalisation of information and content received through digital health technologies has been shown to empower patients to have greater ownership over the management of their health and to contribute towards shifting the power balance from health providers making care decisions on behalf of patients to a shared decision-making process [15,164]. It is thus recommended that a personalised approach is undertaken in the development and operation of digital health technologies as it could be pivotal in increasing patient engagement with their care [15].

Almost all studies included in this review (n = 120) reported predominantly positive appraisals of patient-facing digital technologies in terms of acceptability, feasibility, and/or utility. Since studies were not evaluated for their methodological quality, as this was outside the scope of the present review, any reported findings by individual studies should be interpreted with caution. Nevertheless, these findings echo those of other research, suggesting that cancer patients may be overall amenable to the incorporation of digital health technologies into their care [165,166]. Therefore, future research should direct its efforts into exploring strategies for the successful implementation of digital health technologies into routine cancer care practice, including the identification of both individual- and institutional-level barriers and facilitators to the sustainable long-term use of such technologies in this context. The findings from the present review could contribute to this direction by providing evidence to guide investigations into the technology-associated parameters influencing patients’ decisions regarding the adoption of such technologies.

### 4.2. Strengths and Limitations

A comprehensive approach to locate relevant articles was undertaken in this review, using three databases and a combination of subject headings and free-text terms without applying any date restrictions and performing a rigorous forward and backward citation search for all records meeting the inclusion criteria. Nevertheless, there may have been publications which were missed. The identification of 21 additional eligible publications through citation searching may be suggestive of this. Moreover, the exclusion of grey literature and non-English-language publications may mean that studies providing evidence on cancer patients’ perspectives and requirements for patient-facing digital technologies published outside the traditional academic channels or in languages other than English were also missed. Despite these limitations, a large number of eligible studies were identified and included in this review (n = 128), enabling a rich synthesis of evidence.

In line with the broad scope of this review, studies reporting on cancer patients’ perceptions of digital technologies employed for their care were considered eligible for inclusion regardless of their research design or the methods employed for the assessment of patients’ perceptions. However, more than half of the included studies (n = 68) used either standardised or ad hoc questionnaires as their primary means of evaluating patients’ perceptions of digital health technologies. This means that the technological parameters considered by patients were limited to those included in the measures used by individual studies. It is possible, therefore, that the patient requirements presented in this review do not adequately reflect all technological parameters considered important by cancer patients, and that potentially relevant technological requirements were missed. Furthermore, although no restrictions on cancer type were applied in our search, a large number of the included studies (n = 75) were conducted with breast, prostate, lung, colorectal, and head and neck cancer patients and survivors. Hence, the reported findings may not capture the views of or be applicable to cancer patient and survivor populations outside of those represented in the present review.

## 5. Conclusions

To the authors’ knowledge, the present scoping review is the first to systematically map and synthesise the existing evidence on cancer patients’ perspectives and requirements for patient-facing digital technologies. An increasing number of studies are using digital technologies to support cancer care. These predominantly comprise Web-based software/platforms, mobile or smartphone devices and/or applications, and remote sensing and wearable technologies, mostly employed for the delivery of self-management interventions and the monitoring of patients’ symptoms and physiological parameters. Several technological features and parameters are considered important by cancer patients. Of these, the most commonly reported include the utility of technologies in enabling the management of patients’ symptoms and overall conditions, their user-friendliness, and their ability to facilitate patient–clinician interactions. The findings from this review provide evidence that could inform future research on technology-associated parameters influencing cancer patients’ decisions regarding the uptake and adoption of patient-facing digital health technologies.

## Figures and Tables

**Figure 1 cancers-16-02293-f001:**
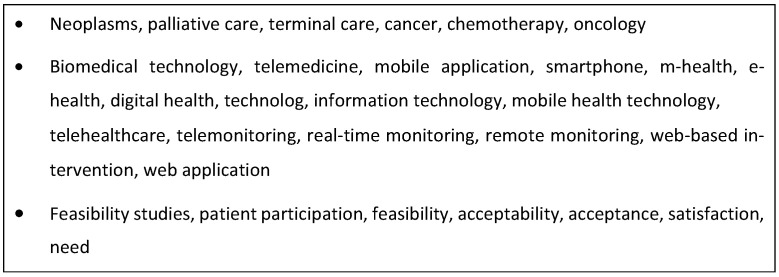
Search terms applied in database searches.

**Figure 2 cancers-16-02293-f002:**
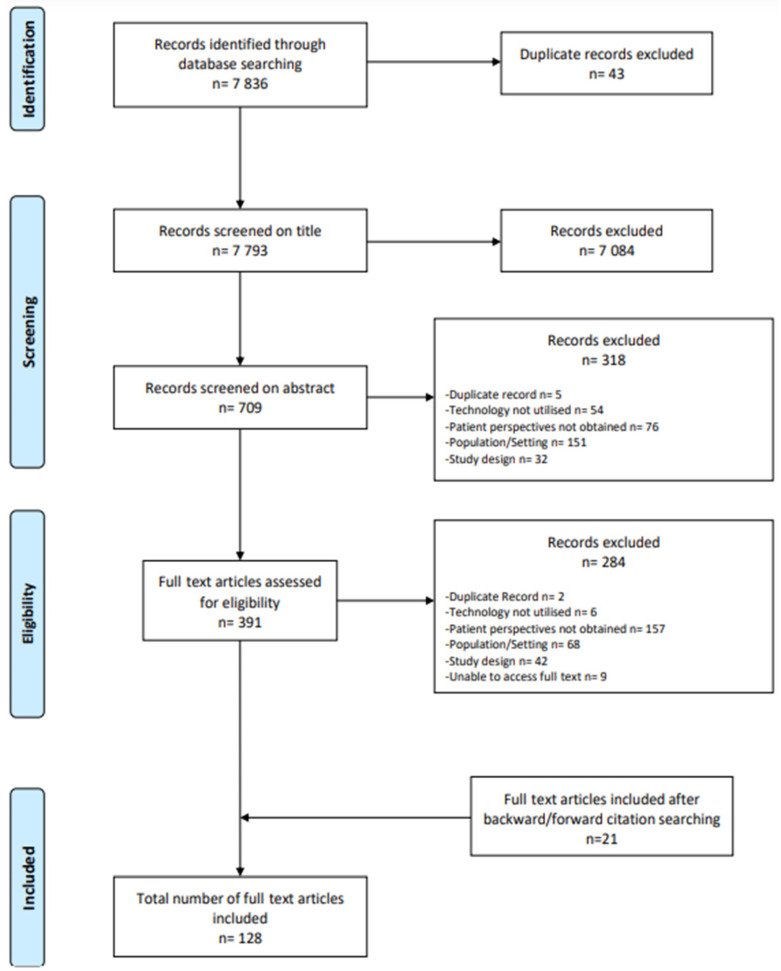
PRISMA flow diagram of the article screening and selection process [23].

**Figure 3 cancers-16-02293-f003:**
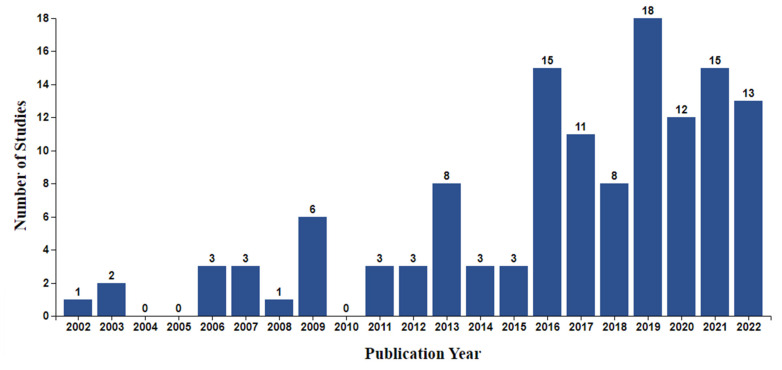
Number of identified studies by publication year.

**Figure 4 cancers-16-02293-f004:**
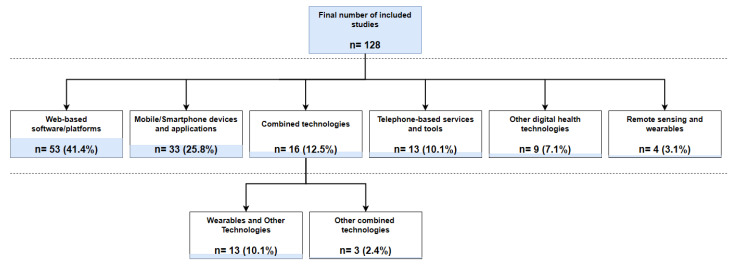
Number of identified studies by technology category.

**Figure 5 cancers-16-02293-f005:**
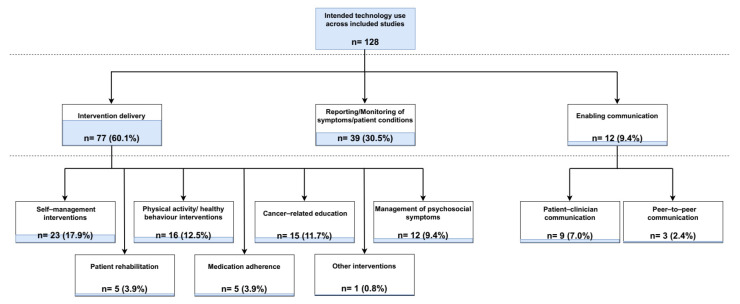
Number of included studies by intended technology use.

**Figure 6 cancers-16-02293-f006:**
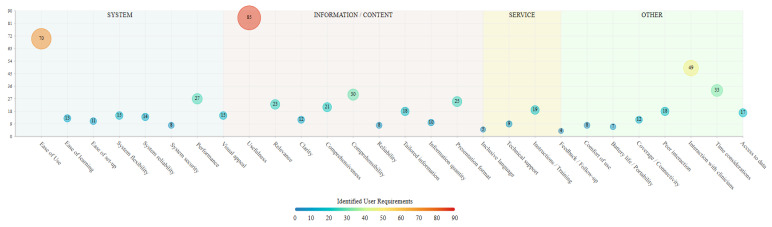
Identified user patient-facing digital health technology requirements by classification category.

**Table 1 cancers-16-02293-t001:** Review question (PICO).

Population	Intervention	Comparison	Outcome
Cancer patients/survivors	Patient-facing digital health technologies	N/A	User perspectives/requirements

N/A: Not applicable.

## Data Availability

The data presented in this study are available in this article.

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
