# Peer review of "Cancer Patients’ Perspectives and Requirements of Digital Health Technologies: A Scoping Literature Review"

_cancers, 2024, doi:10.3390/cancers16132293_

Round 1
Reviewer 1 Report (Previous Reviewer 3)
Comments and Suggestions for Authors
Subject: Peer Review Comments for Manuscript: "Cancer patients’ perspectives and requirements of digital health technologies: A scoping literature review"
This manuscript offers a valuable overview of digital health technologies from the perspective of cancer patients, highlighting critical factors for their adoption and usage. While the topic is timely and the manuscript is generally well-prepared, there are several issues where clarification and improvement are required to enhance the manuscript's quality.
1. Search Strategy (Figure 1): Regarding Figure 1 in your manuscript, it is unclear whether the search strategy was confined to step 4 alone or if it involved executing all steps (1-4) followed by combining the results. To enhance clarity and ensure the reproducibility of your methodology, a detailed explanation of the search strategy and the selection criteria applied is essential. Please consider revising this section to clearly delineate the steps followed during the literature search and study selection process.
2. Completeness of Figure 2: It appears that Figure 2 is incomplete or truncated in the provided manuscript. It is suggested to verify the figure files and ensure that the entire figure is displayed correctly to facilitate complete understanding of the depicted data.
3. Readability of Table 3: The current format of Table 3 hampers readability. Consider reformatting this table or employing graphical elements (e.g., charts or simplified diagrams) to present the data more effectively. Such visual enhancements could significantly aid in conveying complex information succinctly.
4. Visualization of Section 3.5 (User Perspectives and Requirements): The section discussing user perspectives and requirements is critical but is currently presented only in text form, which may be challenging for readers to understand. Incorporating illustrative figures or diagrams that summarize these perspectives and requirements could greatly enhance understanding and engagement with the content.
5. Scope of Included Studies: The review restricts its focus to studies involving patients with breast, prostate, lung, colorectal, and head and neck cancers. While these are prevalent, omitting other common cancers such as liver and gastric cancers may limit the generalizability of the conclusions. I recommend discussing this limitation more explicitly in the manuscript and suggesting areas for future research that include a broader range of cancer types.
Author Response
Reviewer 1
Comment 1: This manuscript offers a valuable overview of digital health technologies from the perspective of cancer patients, highlighting critical factors for their adoption and usage. While the topic is timely and the manuscript is generally well-prepared, there are several issues where clarification and improvement are required to enhance the manuscript's quality.
Response: We thank the reviewer for their feedback. Please find below our responses to each of the comments together with references to corresponding changes in the manuscript.
Comment 2: Search Strategy (Figure 1): Regarding Figure 1 in your manuscript, it is unclear whether the search strategy was confined to step 4 alone or if it involved executing all steps (1-4) followed by combining the results. To enhance clarity and ensure the reproducibility of your methodology, a detailed explanation of the search strategy and the selection criteria applied is essential. Please consider revising this section to clearly delineate the steps followed during the literature search and study selection process.
Response: Figure 1 presents the search strings used for locating relevant records as part of the database searching process discussed in Section 2.2. Sections 2.3 and 2.4 describe the criteria and the steps taken to select eligible records for inclusion in our review, once relevant records were retrieved from the searched databases. We agree that Figure 1 may confuse readers regarding the steps taken to locate eligible studies and we have therefore updated it to present the search terms used, instead of the strategy.
Comment 3: Completeness of Figure 2: It appears that Figure 2 is incomplete or truncated in the provided manuscript. It is suggested to verify the figure files and ensure that the entire figure is displayed correctly to facilitate complete understanding of the depicted data.
Response: We have now updated Figure 2 to address this.
Comment 4: Readability of Table 3: The current format of Table 3 hampers readability. Consider reformatting this table or employing graphical elements (e.g., charts or simplified diagrams) to present the data more effectively. Such visual enhancements could significantly aid in conveying complex information succinctly.
Response: Thank you for this point. We have added a figure (Figure 6 in the manuscript) to summarise the information provided in Table 3 and aid its comprehensibility.
Comment 5: Visualization of Section 3.5 (User Perspectives and Requirements): The section discussing user perspectives and requirements is critical but is currently presented only in text form, which may be challenging for readers to understand. Incorporating illustrative figures or diagrams that summarize these perspectives and requirements could greatly enhance understanding and engagement with the content.
Response: We thank the reviewer for this suggestion. We have created a figure to map the information discussed in section 3.5 (Figure 6 in the manuscript) to enable a more comprehensive and accessible presentation of this information.
Comment 6: Scope of Included Studies: The review restricts its focus to studies involving patients with breast, prostate, lung, colorectal, and head and neck cancers. While these are prevalent, omitting other common cancers such as liver and gastric cancers may limit the generalizability of the conclusions. I recommend discussing this limitation more explicitly in the manuscript and suggesting areas for future research that include a broader range of cancer types.
Response: We agree with the reviewer that although no cancer type restrictions were applied in our search, a large proportion of eligible studies were conducted with patients diagnosed with the abovementioned cancer types and therefore our findings may not be directly generalisable or transferable to other cancer populations. We have now discussed this limitation in the ‘Strengths and limitations’ section of the manuscript (lines 467-472).
Reviewer 2 Report (Previous Reviewer 2)
Comments and Suggestions for Authors
I accept the responses from the authors as per my concerns.
Comments on the Quality of English LanguageNo comment.
Author Response
Reviewer 2
Comment: I accept the responses from the authors as per my concerns.
Response: We thank the reviewer for their time and constructive feedback to our manuscript.
Reviewer 3 Report (Previous Reviewer 1)
Comments and Suggestions for Authors
This literature review aimed to investigate cancer patients’ perspectives and requirements of digital health technologies. The authors made efforts to collect as many studies as they could.
My main concerns are as follows:
1: A) The authors did not provide enough study details or characteristics. Table 2 should have included more details about study populations such as age, sex, country, and certain demographic criteria. B) Besides, the technology used should have been described. For example, it is not enough to mention a web-based program. The authors should have described this program. Without such details, the SR will turn into a reference-section paper as readers have to go to the cited paper every time they need to get simple information. C) More importantly, the authors did not provide any details about the results of the studies.
2: Quality assessment and bias detection are among the main components of SRs. The authors did not assess the quality and risk of bias of retrieved articles.
3: The authors concluded that patient-facing digital technologies were found to be acceptable, feasible, and useful to cancer patients when meeting their needs. How could they reach such findings without investigating the results of the retrieved articles?
Author Response
Reviewer 3
Comment 1: This literature review aimed to investigate cancer patients’ perspectives and requirements of digital health technologies. The authors made efforts to collect as many studies as they could. My main concerns are as follows: A) The authors did not provide enough study details or characteristics. Table 2 should have included more details about study populations such as age, sex, country, and certain demographic criteria. B) Besides, the technology used should have been described. For example, it is not enough to mention a web-based program. The authors should have described this program. Without such details, the SR will turn into a reference-section paper as readers have to go to the cited paper every time they need to get simple information. C) More importantly, the authors did not provide any details about the results of the studies.
Response: We thank the reviewer for these points. We agree that the manuscript would benefit from including additional information on the identified technologies as well as the countries in which included studies were conducted. We have now extracted this data and included it in Table 2 (please see columns ‘Country’ and ‘Description of technology’), and added a description of the extracted country information in the ‘Results’ section (lines 167-168).
The main focus of our review was on cancer patients’ requirements of digital health technologies. Therefore, the data extracted from individual studies and synthesised in the manuscript centre on the identified technologies and patients’ requirements of these. It is our belief that presenting additional information which is not closely linked to the review’s objectives is outside the scope of the present review and may detract readers from the overall purpose of our paper.
Comment 2: Quality assessment and bias detection are among the main components of SRs. The authors did not assess the quality and risk of bias of retrieved articles.
Response: We believe that our review can be best classified as a scoping review as its main objective was to identify and map existing evidence on cancer patients’ requirements of patient-facing digital health technologies. A risk of bias assessment/critical appraisal is not recommended for scoping reviews according to the PRISMA Extension for Scoping Reviews (PRISMA-ScR) and other key guidelines for the conduct and reporting of scoping reviews (e.g., Peters et al. 2020, Updated methodological guidance for the conduct of scoping reviews), and was therefore not performed in our review. We hope that our rationale for not undertaking a risk of bias assessment is now clearer to the reviewer.
Comment 3: The authors concluded that patient-facing digital technologies were found to be acceptable, feasible, and useful to cancer patients when meeting their needs. How could they reach such findings without investigating the results of the retrieved articles?
Response: We thank the reviewer for this comment. We agree that this part of the conclusions may not be well supported, given that evidence on patients’ perceptions of the acceptability, feasibility and usefulness of technologies were not critically appraised. We have therefore removed this section from the conclusions.
Round 2
Reviewer 3 Report (Previous Reviewer 1)
Comments and Suggestions for Authors
No more comments.
This manuscript is a resubmission of an earlier submission. The following is a list of the peer review reports and author responses from that submission.
Round 1
Reviewer 1 Report
Comments and Suggestions for Authors
This study aimed to investigate the perspectives of cancer patients regarding digital health technologies. I have the following comments:
1: Since studies were systematically retrieved, this paper is a systematic review. This should be clarified across the manuscript.
2: The main findings of the retrieved papers should be added to Table 1, mentioned in the results section, and discussed in the discussion section.
3: The quality of retrieved studies should be assessed to determine the overall quality of evidence.
4: Tables and figures need to be edited because their size is bigger than the file.
5: Ref 7: The name of the journal was not included in many references (7 and 20, for example), and the abbreviations of the journal names were not written in many other cases. the reference section needs extensive editing.
Comments on the Quality of English LanguageNo big issues were detected.
Author Response
Reviewer 1
Comment: Since studies were systematically retrieved, this paper is a systematic review. This should be clarified across the manuscript.
Response: We thank the reviewer for this comment. The PRISMA Extension for Scoping Reviews (PRISMA-ScR) and other key guidelines for the conduct and reporting of scoping reviews (e.g., Peters et al. 2020 Updated methodological guidance for the conduct of scoping reviews), describe the systematic process in which evidence ought to be identified when undertaking a scoping review. In line with these recommendations, we followed a systematic approach to locate and retrieve eligible studies in our review.
Comment: The main findings of the retrieved papers should be added to Table 1, mentioned in the results section, and discussed in the discussion section.
Response: We acknowledge the issue with the format conversion during the initial submission of our manuscript which resulted in tables and figures being displaced from their intended positions. We have since addressed this issue, ensuring that Table 1 now accurately reflects the main findings of the retrieved papers. Additionally, we have incorporated references to Table 1 within the results section to provide readers with a clear overview of key findings. Furthermore, we have elaborated on the implications of these findings in the discussion section of the manuscript. We believe that these revisions have strengthened the clarity and comprehensiveness of our manuscript.
Comment: The quality of retrieved studies should be assessed to determine the overall quality of evidence.
Response: We believe that our review can be best classified as a scoping review as its main objective was to identify and map existing evidence on cancer patients’ perspectives of patient-facing digital health technologies. A risk of bias assessment is not recommended for scoping reviews according to the PRISMA Extension for Scoping Reviews (PRISMA-ScR), and was therefore not performed in our review. We hope that our rationale for undertaking a scoping review is now clearer to the reviewer.
Comment: Tables and figures need to be edited because their size is bigger than the file.
Response: We apologise for the inconvenience caused by the oversized tables and figures in the initial submission, since during the system conversion the margins were altered. We have taken your feedback into consideration and have made the necessary adjustments to resize the tables and figures to ensure they comply with the file size requirements of the journal. We trust that these modifications have addressed the concerns raised and improved the readability and accessibility of our manuscript.
Comment: Ref 7: The name of the journal was not included in many references (7 and 20, for example), and the abbreviations of the journal names were not written in many other cases. the reference section needs extensive editing.
Response: We thank the reviewer for spotting these inconsistencies. We have now updated the reference section to rectify these.
Reviewer 2 Report
Comments and Suggestions for Authors
This systematic review focuses on the utilization of digital health technologies in cancer therapy. Upon reviewing the manuscript, I find myself puzzled regarding the scope of this review paper in relation to the term "digital health technology." Digital technology in healthcare encompasses a wide array of technologies, including electronic health records, remote patient monitoring, digital therapeutics, health chatbots and virtual assistants, blockchain in healthcare, 3D printing in healthcare, and AI in medical imaging. While the authors have incorporated some of these technologies into their review, there are several others that are conspicuously absent but crucial to mention. Additionally, Table 1 appears overly simplistic, Figure 1 lacks clarity as a figure, and Tables 2 and 3 are incomplete, further detracting from the comprehensiveness of the paper.
Comments on the Quality of English LanguageNo problem.
Author Response
Comment: This systematic review focuses on the utilization of digital health technologies in cancer therapy. Upon reviewing the manuscript, I find myself puzzled regarding the scope of this review paper in relation to the term "digital health technology." Digital technology in healthcare encompasses a wide array of technologies, including electronic health records, remote patient monitoring, digital therapeutics, health chatbots and virtual assistants, blockchain in healthcare, 3D printing in healthcare, and AI in medical imaging. While the authors have incorporated some of these technologies into their review, there are several others that are conspicuously absent but crucial to mention.
Response: We thank the reviewer for this insightful comment. We previously stated across different parts of our manuscript, including the review question and eligibility criteria sections, that our review is looking at ‘patient-based’ or ‘patient-centred’ digital health technologies to describe digital technologies with which patients interact to participate in health care or clinical activities. We understand that these terms may be confusing and have therefore opted to use the term ‘patient-facing’ instead which provides a more accurate description of the reviewed technologies. We have now provided a definition for patient-facing digital health technologies (lines 90-92, p. 2) and have added the term ‘patient-facing’ across the manuscript (please see highlighted text in various parts of the manuscript).
Moreover, we have only included studies which reported on patients’ perspectives of patient-facing digital technologies (as stated in our inclusion criteria), and so the presented technologies are the ones for which we could locate available evidence based on our eligibility criteria.
Comment: Additionally, Table 1 appears overly simplistic, Figure 1 lacks clarity as a figure, and Tables 2 and 3 are incomplete, further detracting from the comprehensiveness of the paper.
Response: We acknowledge the feedback regarding the visual elements of our manuscript, specifically Table 1 and Figure 1. During the initial submission, format conversion issues resulted in Table 1 appearing overly simplistic, and Figure 1 lacked the necessary clarity to effectively convey its intended message. We also recognise that Tables 2 and 3 were incomplete due to formatting issues, which detracted from the comprehensiveness of the paper. We have now revised Table 1 to provide a more comprehensive overview of the main findings of the retrieved papers. We have also made significant improvements to Figure 1, ensuring that it now effectively communicates the relevant information. Furthermore, we have rectified the issues with Tables 2 and 3, ensuring that they accurately represent the data.
Reviewer 3 Report
Comments and Suggestions for Authors
There are plenty of issues in the structure of this manuscript. Like Figure 2, Figure 4, Figure 5, Table 1. It is not very well-organized when converting word version to PDF version, making some information missing in figures and table. Therefore, I think it is not appropriate to publish this paper to Cancers.
Author Response
Comment: There are plenty of issues in the structure of this manuscript. Like Figure 2, Figure 4, Figure 5, Table 1. It is not very well-organized when converting word version to PDF version, making some information missing in figures and table. Therefore, I think it is not appropriate to publish this paper to Cancers.
Response: We thank the reviewer for their feedback. We sincerely apologize for the issues encountered with the structure and presentation of our manuscript, particularly with Figures 2, 4, 5, and Table 1. We acknowledge that the conversion from the Word to the PDF version in the journal’s platform resulted in missing information in these visual elements, which has impacted the clarity and comprehensiveness of the manuscript. We have reviewed and edited the manuscript to ensure that all figures and tables accurately represent the data and findings of our study. Additionally, we have taken steps to enhance the overall organisation of the manuscript to improve its readability and coherence. We hope that the revised version of our manuscript meets the standards of Cancers.